# Lotka's Wheel and the long arm of history: how does the distant past determine today's global rate of energy consumption?

Timothy J. Garrett[1,*], Matheus R. Grasselli[2], and Stephen Keen[3]

[1]University of Utah, Department of Atmospheric Sciences, 135 S 1460 E, Rm 819, Salt Lake City, Utah, 84112, United States
[2]McMaster University, Department of Mathematics and Statistics, Hamilton, ON L8S 4K1, Canada
[3]University College London, London, WC1E 6BT, United Kingdom

**Correspondence:** Tim Garrett (tim.garrett@utah.edu)

**Abstract.** Global economic production – the world GDP – has been rising steadily relative to global primary energy demands, lending hope that technology can drive a gradual decoupling of society from its resource needs and associated environmental pollution. Here we present a contrasting argument: in each of the 50 years following 1970 for which reliable data are available, one Exajoule of world energy was required to sustain each $5.50 \pm 0.21$ trillion year-2019 US dollars of a global wealth quantity defined as the cumulative inflation-adjusted economic production summed over all history. No similar scaling was found to apply between energy consumption and the more familiar quantities of yearly economic production, capital formation, or physical capital. Considering the scaling held over half-a-century, a period that covers two thirds of the historical growth in world energy demands, the implication is that inertia plays a far more dominant role guiding societal trajectories than has generally been permitted in macro-economics models, or by policies that prescribe rapid climate mitigation strategies. Rather, environmental impacts will remain strongly tethered to even quite distant past economic production – an unchangeable quantity. As for the current economy, it will not in fact decouple from its resource needs. Instead, simply maintaining existing levels of world inflation-adjusted economic production will require sustaining growth of energy consumption at current rates.

## 1 Introduction

Alfred J. Lotka regarded the "life-struggle" as a competition for available energy. The role in this struggle of any physical system, subject to external constraints, is to maximize the flow of energy through it. Lotka proposed, "The influence of man, as the most successful species in the competitive struggle, seems to have been to accelerate the circulation of matter through the life cycle, both by 'enlarging the wheel', and by causing it to "spin faster"... the physical quantity in question is of the dimensions of power". "In every instance considered, natural selection will so operate as to increase the total mass of the organic system, to increase the rate of circulation of matter through the system, and to increase the total energy flux through the system, *so long as there is presented an un-utilized residue of matter and available energy.*" (Lotka, 1922) (our italics).

Adopting Lotka's perspective, as illustrated in Fig. 1, the field of thermodynamics should be seen as essential to any understanding or treatment of societal actions. Yet, even a century later, its consideration remains a fringe view, even in the economic treatments most widely used to guide economic and climate policy (Tol, 2018; Nordhaus, 2017). "Production functions" treat resource extraction as just one sector of the economy, no more significant than, for instance, the services sector. These modeling

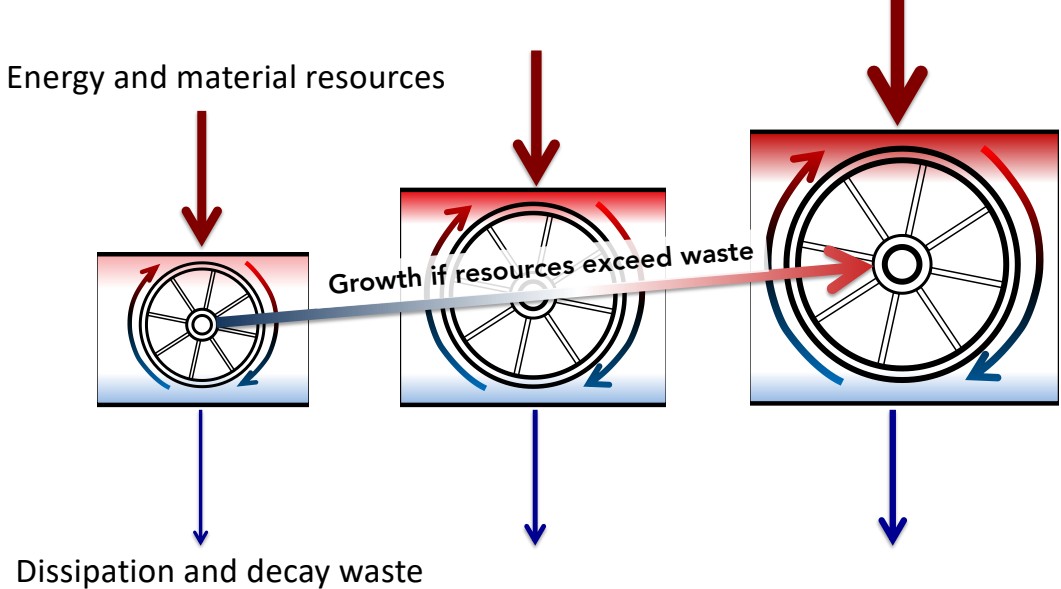

Energy and material resources

Growth if resources exceed waste

Dissipation and decay waste

**Figure 1.** Representation of Lotka's view on the thermodynamic mechanisms governing system growth, involving a wheel that enlarges and accelerates using an "un-utilized residue" of energy and matter representing the difference between consumed resources and waste.

frameworks permit improvements to technology and efficiency as key policy tools for simultaneously lifting human prosperity while limiting adverse impacts from resource depletion and environmental degradation through waste production (Victor, 2010; Deutch, 2017).

As a counterpoint to the traditional approach, our past work described a new macroeconomic quantity – historically cumulative production – that we demonstrated to have had a quantifiable constant relationship with world primary energy resource demands, or civilization's collective power. A consequence of the relationship is that the inflation-adjusted GDP is more closely related to a surplus of energy – or Lotka's "un-utilized residue" – than to the current rate of energy consumption itself (Garrett, 2011, 2012; Garrett et al., 2020). Here, we use a longer available data set to show that the relationship has held for a half-century, covering the period between 1970 and 2019. This new time series of historically cumulative production suggests a "top-down" metric for facilitating discussions of what is possible in hypothetical scenarios of future interactions between society, natural resource availability, and climate change.

## 2  A scaling between energy consumption and historically cumulative production

To avoid complications associated with the details of trade, interactions between economic sectors, or distinctions between energy types, this study is focused only on global quantities, as described in the Materials and Methods below. Annual primary energy sources, those that are available to drive civilization activities of whatever type, are consumed and ultimately dissipated as waste heat at a rate that can be expressed as an instantaneous quantity $E$ (e.g., Terawatts) or a yearly-averaged quantity $E_i$

with units of power (e.g., either Terawatts or Exajoules per year) (Garrett et al., 2020). For example, $E_{2019} = 609$ means that humanity during the course of 2019 was powered by 609 Exajoules or at a rate of 19.3 Terawatts. Annual economic production (Gross Domestic Product) or output is defined monetarily as the sum of tallied financial exchanges made to acquire final goods and services within a given year. After adjusting for inflation, we denote this quantity as $Y_i$, expressed in units of constant 2019 USD per year, effectively a yearly average of the instantaneous rate $Y$ in 2019 USD per year.

Given that humanity's billions emerged from the past, the magnitude of civilization's annual energy demands might be thought to be tied to an economic quantity that is not a rate – as it is for $Y$ – but rather one that has accumulated through time and has units of currency. The first candidate we consider for such an accumulated quantity is economic capital $K_i$, a primary factor in traditional models of economic production. The second is a new quantity, the time integral of production, not just over one year – as is done in calculation of $Y_i$ – but over the entirety of history, what we term the world historically cumulative production $W_i = \sum_{j=1}^{i} Y_j$. Expressed in continuous form

$$W(t) = \int_0^t Y(t') \, dt' \tag{1}$$

The contribution of depreciation and decay to $W$ is addressed later.

Time series for $Y_i$, $K_i$, $W_i$ and $E_i$ are shown in Figure 2 for a 50 year period between 1970 and 2019. Global energy consumption $E$ increased by a factor of 2.8, production $Y$ increased by a factor of 4.5, and economic capital $K$ increased by a factor of 7.9. A related quantity, the rate of capital formation, $dK/dt$, is not shown because it is implicit in the curve for $K$, however, as is evident for the curve for $K$, its value varied considerably. While the ratio $(dK/dt)/E$ increased by a factor of 1.5 between 1970 and 2019, the relative increase was 3.2 in 2009 and 0.34 in 1982. The ratio $y = Y/E$, sometimes termed the energy productivity, trended steadily upward.

Defining growth rates in quantity $X$ as $R_X = (1/X)dX/dt = d\ln X/dt$, a least-squares fit to the data gives $R_y = 1.00\%$ per year. Meanwhile, the ratio $k = K/E$ grew at rate $R_k = 1.96\%$ per year, nearly twice as fast as $y$, or a doubling time of 35 years. The appealing picture presented is of an economy that has become rapidly less energy intensive, with technological innovation enabling more to be done with less (Sorrell, 2014).

So it would be natural to infer from a history of increasing $Y/E$ that our human acumen for invention has been driving a long-term decoupling of the global economy from resource constraints. However, comparing $W_i$ and $E_i$ suggests otherwise. Cumulative production $W_i$ increased more slowly than $Y_i$ or $K_i$, by a factor of 2.7 over the 50-year period. This ratio is nearly identical to the factor 2.8 increase found for $E_i$. Expressed (for simplicity) as a continuous function, the ratio $w = W/E$ has fluctuated to some degree, but the average tendency was $R_w = -0.02\%$ per year, far less than the tendencies for either $Y/E$ or $K/E$. The average value is:

$$w = \frac{W}{E} = 5.50 \pm 0.21 \tag{2}$$

in units of trillion 2019 USD of cumulative production per Exajoule of energy consumed each year.

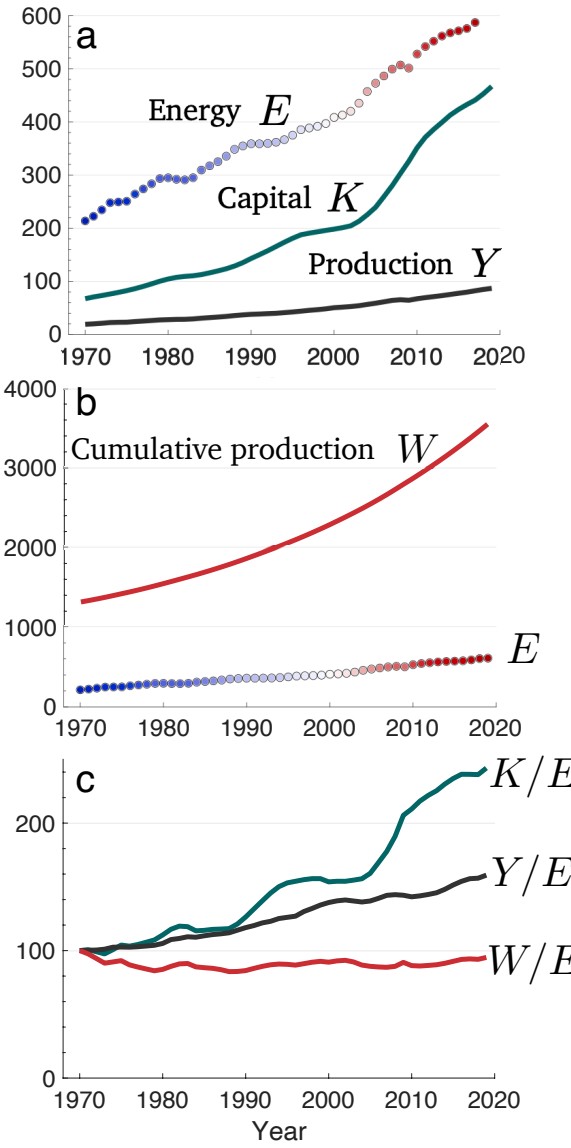

**Figure 2.** a: Time series for the period 1970 to 2019 of global yearly annual primary energy consumption $E_i$ in Exajoules per year, the world annual GDP $Y_i$ in yearly currency, and the total value of physical capital capital stock $K_i$ in units of currency. b: Energy in Exajoules per year and historically cumulative production $W_i$ in currency. c: The ratio of economic values to annual energy consumption, setting the ratio in 1970 to 100. All currency units are in trillion 2019 USD

Considering that the ratio $w$ is nearly a constant, the relationship between $W$ and $E$ does not appear to be one merely of correlation between two quantities, as for example has been noted for $E$ and $Y$ (Jarvis, 2018). Instead $W$ and $E$ have maintained a linear scaling over the half century period for which widely published data are available. A least-squares fit to the logarithms of $W$ and $E$ yields the relationship $W = 5.47E^{1.00}$. Calculated instead as a linear fit, the relevant expression is $W = 5.67E - 66$. Note the intercept of the fit where $E = 0$ is equivalent to $W = -66$ trillion 2019 USD, a value that is just -1.9% of the 2019 value for $W_i$ of 3547 trillion 2019 USD, and so sufficiently small as to plausibly approximate the origin. By contrast, the linear fit for world GDP and energy is $Y = 0.17E - 21$ with an intercept of $Y = -21$ trillion 2019 USD, or -25% of its 2019 value. So, while $Y$ and $E$ may be correlated, they do not scale as in the same manner as $W$ and $E$.

## 3  A production relation

We interpret the quantity identified here as the historically cumulative global production $W$ as an economic expression of the rotational power of Lotka's Wheel, that is the capacity to drive the collective to-and-from of civilization's circulations through the relationship $W = wE$, where $w$ is nearly a constant. Certainly, an objection might be raised that the past 50 years is too short relative to the time span of humanity to draw meaningful conclusions about the relationship of historically cumulative production to current energy demands. Measured in units of years, this may be true. However, the last half-century covers a remarkable two-thirds of humanity's total growth expressed in terms of energy consumption, or 1.5 doublings in $E$, during which a great deal changed in humanity's social and technological makeup.

Taking the first derivative of Eq. 1 yields an inflation-adjusted economic production relation. Assuming $W = wE$ for constant $w$, then

$$Y = \frac{dW}{dt} = w\frac{dE}{dt} \tag{3}$$

Real economic production is related to the *rate of increase* in world primary energy consumption. The implication is that the real GDP is a tally of the instantaneous monetary exchanges that, directly or indirectly, increase civilization's ability to access more energy in the future. For the case that $dY/dt = 0$, namely that there is constant inflation-adjusted economic production $Y$ or zero GDP growth, energy demands expand at rate $Y/w$. If there is GDP growth, as preferred by governments, and $d\ln Y/dt > 0$, then world energy consumption accelerates.

Eq. 3 assumes only that $w$ is a constant, a result that can be readily refuted, or supported as here with decades of data from multiple sources. The approach does nonetheless have some important limitations, notably an inability to resolve short-term, fine-scale behaviors. The evolution of cumulative inflation-adjusted world economic production $W$ is highly smoothed because it is a summation, or integration, over history and the global economy. Even given a strong multi-decadal relationship of $E$ to $W$, year-to-year variability in $E$, such as during recessions or pandemics, cannot be easily related to yearly economic production, especially on national or sectoral scales much smaller than the world as a whole. That said, calculated as a running decadal mean, the average ratio of global production to yearly changes in energy consumption is

$$\widehat{w} = \frac{Y}{dE/dt} = 5.9 \pm 2.2 \tag{4}$$

in units of trillion 2019 USD per Exajoule consumed each year, which is very similar to that expressed for $w$ given by Eq. 2, although the variability is higher given the comparison of $Y$ to a differential in $E$.

Nonetheless, Eq. 3 can also be seen as being highly counterintuitive, as it suggests for the hypothetical limiting case of $dE/dt = 0$ – one where the world attains a sort of metabolic steady-state with energetic and material inputs and outputs in balance – that *real* world economic production disappears, that is $Y = 0$. Such a result would seem highly peculiar viewed from any traditional economic perspective.

It is important to note, however, that zero real, inflation-adjusted production does not forbid non-zero, positive nominal production. If there is a large difference between the nominal and real GDP, it appears in economic accounts as high values of the GDP deflator, or as hyper-inflation. Interpreted physically, civilization dissipates energy along previously produced networks. Even as current production continues to grow these networks, there is concurrent fraying of those previously constructed, sufficient to offset any productive gains Garrett (2014).

A metabolic steady-state may only represent a temporary marker prior to more complete collapse, thermodynamic as well as economic, given the severe constraints to modern society. Along the pathway of contraction, any external resources that become available to civilization would no longer be sufficient to count as an un-utilized residue. Like a patient consumed by cancer, any growth would be more than offset by internal consumption – burning the furniture to heat the house, so to speak. Nominal production might remain, but it would be fueled more by internal than external resources. Eventually, civilization attains the point of complete collapse, whereupon both civilization power and nominal production equal zero.

Certainly there are other macro-economic treatments that consider societal energy demands, although the production functions in these models tend to be highly complex, failing to appeal foremost to the dimensions of the problem. Rather than starting with the constraint that the factors of economic production, of whatever combination, must tally dimensionally to units of currency per time, quantities such as dimensionless capital, labor, and useful work are set to non-integer exponents, or are themselves placed in exponents (Ayres et al., 2003; Ayres and Warr, 2009; Lindenberger and Kümmel, 2011; Keen et al., 2019). Such functions can be shown to reproduce past behaviors for specific nations, but only by way of specifying coefficients, or "output elasticities", that are themselves determined from past economic conditions, and that are allowed to vary according to the time period considered. The production functions are effectively moving targets that can be tuned to accurately reproduce past conditions. They cannot be presumed to express anything fundamental about the long-run evolution of the future. As attributed by E. Fermi to J. von Neumman "with four parameters I can fit an elephant, and with five I can make him wiggle his trunk."

The approach described here is more strictly thermodynamic, and so does not allow for such mathematical flexibility. The collective societal assessment of the final inflation-adjusted value of goods and services $Y$ appears to correspond with "enlarging the wheel" or enabling it to "spin faster", that is the technological innovation of a larger human system, one that is newly consumptive of primary reserves over and beyond the scenario where energy consumption rates stay constant. Current energy demands sustain the wheel's rotation against energy dissipation and material decay. It is only with an excess or "un-utilized residue" of available energy that an effective phase change becomes possible whereby raw materials are converted through eco-

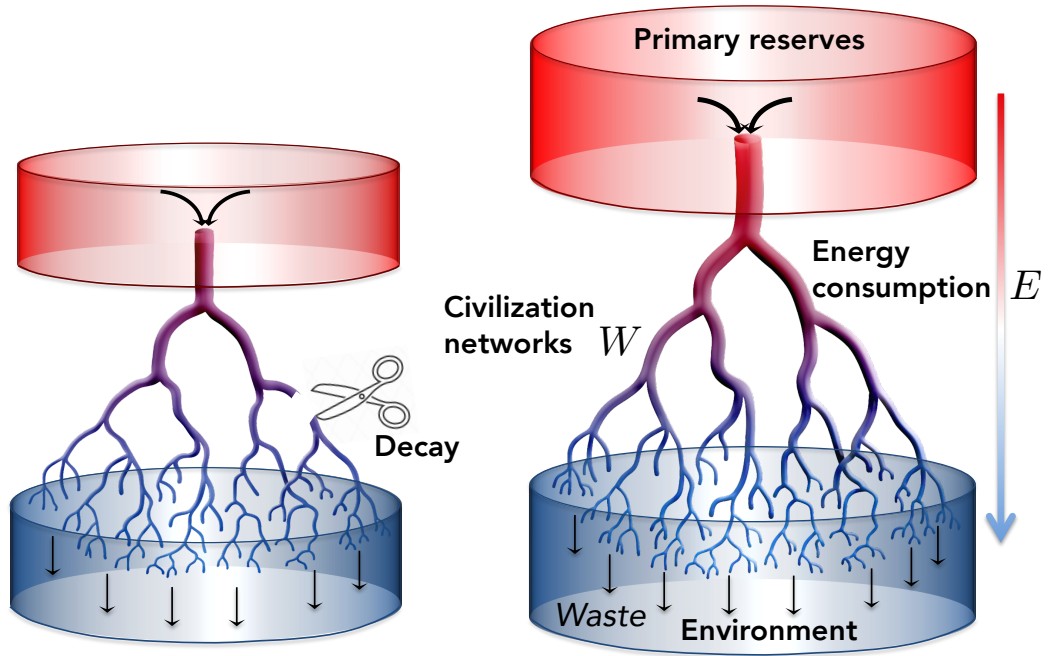

**Figure 3.** Elaboration on Lotka's Wheel. Civilization growth related to increases in its power at rate $dE/dt$ as tied to network production through the inflation-adjusted GDP $Y$. Current power $E$ is thus tied to the historically cumulative GDP through $W = \int_0^t Y(t)\, dt' = w \int_0^t (dE/dt)\, dt' = wE$.

nomic production into newly created civilization networks. With increasing available energy and power, adjusting for network decay, societal movements are accelerated along these enlarged networks. (Figure 3).

In fact, there is some evidence that civilization size and speed are two independent modes of variability whose rates of change are nearly equally divided. A linear scaling has been noted between the size of a city's population and how fast its inhabitants walk (Bettencourt et al., 2007). More globally, over the 50 year period considered, world population – as a measure of size – increased at an average rate of 1.46% per year. Meanwhile, per capita world GDP – as a plausible metric for speed – increased at the nearly equivalent rate of 1.55% per year.

## 4  Contributions of the distant past to the present

At some level, the empirical nature of Eq. 2 stands on its own, and so too its implications for economic production through Eq. 3. Its essence is that current civilization value $W$ and energy consumption $E$ is not a direct consequence of current economic transactions, but instead reflects a historical pathway. By way of analogy, consider the circulations within our bodies, brains, and machines, and our activities such as housework, transport to and from work and the grocery store, and even conversation

among family and friends, that all of these require current energy consumption in some form. Each one of these may involve a financial transaction at some prior stage, for cleaning products, gasoline, or food, but crucially no financially quantifiable purchase is made at the point at which the energy is consumed, only in the past.

Some might counter that economic models already account for recent purchases, or that historically distant production and consumption cannot linger to contribute to energy demands today. Fig trees grown for the enjoyment of Ancient Greeks would seemingly have nothing to do with the power consumption of internet servers today.

Such an argument about the importance of the past can be tested. The effective lifetime of prior production can be easily estimated for those models that employ traditional economic accounting. Capital is formed through economic production $Y$ after subtracting both depreciation at rate $\delta$ and consumption $C$ of goods and services. The underlying equation is $dK/dt = (Y - C) - \delta K$. Expressing consumption as $C = cY$ and adopting a simplified production function of form $Y = \beta K$ where $\beta = Y/K$ is the production efficiency (or the inverse of the capital-to-output ratio), it follows that the rate of capital formation is

$$\frac{dK}{dt} = (\beta - \delta')K, \tag{5}$$

Dividing both sides by $K$, the exponential growth rate of capital is $R_K = (\beta - \delta')$ where $\delta' = \delta + c\beta$. Purely mathematically speaking, consumption itself can be viewed as a form of depreciation of very short-lived capital at rate $c\beta$, in addition to depreciation at rate $\delta$.

The value of the modified depreciation term $\delta'$ can be obtained using data for $Y_i$ and $K_i$. The value of $\beta = Y/K$ over the past 50 years has slowly declined at an average rate of 0.95% per year. Its average value was approximately 0.24 or 24%. Meanwhile, capital grew at an average annual rate of 4.0%. So, the implication is that the annual rate $\delta' = R_K - \beta$ of capital devaluation that owes to combined consumption and depreciation is approximately $24\% - 4\% = 20\%$. Effectively, traditional economic growth models imply that previously produced capital halves its value within just 3.5 years.

Well-known concerns may be raised about any comparison of rates of capital formation with capital valuation, and with how valuations of varied capital stocks should be aggregated (Samuelson, 1966; Sraffa, 1975). Nonetheless, whatever the uncertainties, this inference that traditional economic models see capital $K$ that halves its value in just 3.5 years seems preposterous. The benefits of past productivity clearly persist for much longer. We may no longer use the personal computers of the 1980s, but we would not have current devices without that seminal transformation. Going back further, Ancient Greek fig trees died over 2000 years ago, but many important aspects of the culture of fig-eating Ancient Greeks continue to today.

The crux of this historical valuation problem is critical for judgments of the value of economic models for predicting our future. It appears that the long-distant, or even fairly recent contributions of humanity to politics, science, athletics, architecture, and language are implicitly ignored in traditional economic accounting. Perhaps this is simply because historically important innovations – such as controlled combustion, or the alphabet – cannot be monetized on the open market, even though without them most of modern infrastructure for wealth-generation would collapse. Like "dark-matter" in astronomy that cannot be seen but is known to be the bulk of our universe, there appears also to be a "dark-value" in economics.

T. Piketty describes the issue well: "All wealth creation depends on the social division of labor and on the intellectual capital accumulated over the entire course of human history," continuing "the total value of public and private capital, evaluated in terms of market prices for national accounting purposes, constitutes only a tiny part of what humanity actually values - namely, the part that the community had chosen (rightly or wrongly) to exploit through economic transactions in the marketplace" (Piketty, 2020).

The contribution of the distant past points to the critical importance of considering societal inertia. Here, we showed that historically cumulative production $W$ is a full order of magnitude larger than capital $K$ as valued by current markets, and so should be expected to be equally less resistant to change. The finding that $W$ and $E$ maintain a fixed scaling is thus important as it indicates that energy consumption is required not just to sustain that which we believe potentially available to be sold today – that which loses value within years – but also the unspoken "dark-value" of that which was previously produced, forming the foundations of human culture, and cannot be easily erased.

There are important analogs in the biological and physical world that may provide a useful guide to economic growth theory. For the analogy of Lotka's Wheel, the energy of rotation is the product of its mass and the square of its radius and rotational frequency, all quantities that increase through a cumulative history of positive material and energetic increments. In a cloud, a snow crystal grows through the diffusion of vapor molecules; current vapor consumption depends on the reach of the crystal branches into the surrounding vapor field, insofar as the branches have built upon a prior accumulation of condensed vapor residing within the unexposed crystal interior (Lamb and Verlinde, 2011). The leaves of a deciduous tree enable photosynthesis that fuels fluid circulations through the exterior sapwood; the leaves die seasonally as the sapwood turns into heartwood that, while not actively connected to a larger rejuvenated leaf crown in the following year, structurally supports it (Shinozaki et al., 1964; Oohata and Shinozaki, 1979). Systems may even undergo quite dramatic changes in character while maintaining at all stages a dependence on previously consumptive states, such as with the succession of species that occurs during development of new forest following a major disturbance (Oliver, 1980). Inevitably growth includes loss, through friction for a wheel, evaporation or breakup for a snow crystal, and disease and predation for a tree or forest. But, in all cases, historical past consumption is the primary determinant of the system's current energetic demands.

More sophisticated treatments of civilization's growth trajectory consider the size of the interface that separates it from its surroundings, and how that interface evolves through resource discovery and environmental decay. The resulting dynamic equations are fundamentally logistic in nature, that is they exhibit an exponential response to resource discovery followed by saturation or diminishing returns. Defined as an initial value problem, they can be shown to accurately hindcast the evolution of energy consumption and GDP growth for a period covering 1960 to 2010 (Garrett, 2014, 2015).

**Conclusions**

We have identified a nearly constant value $w$ relating world historically cumulative inflation-adjusted economic production $W$ and current energy demands $E$. The scaling $W = wE$ has held for the past half-century, a period during which resource consumptive demands nearly tripled, suggesting that humanity's current metabolic needs are best considered as emerging

from past innovations that allowed for surplus Haff (2014); Garrett et al. (2020). The relationship's persistence appears to place substantial bounds on humanity's future interactions with its environment. It implies that present sustenance cannot be decoupled from past growth, or that inertia plays a much greater role in societal trajectories than has been broadly assumed, especially in the integrated assessment models widely used to evaluate the coupling between humanity and climate (Nordhaus, 2017).

Thus, even if world GDP growth falls to zero from its recent levels close to 3% per year, long-term decadal-scale growth in resource demands and waste production will continue to accelerate. It is only by collapsing the historic accumulation of wealth we enjoy today, effectively by shrinking and slowing Lotka's Wheel, will our resource demands and waste production decline. Eq. 1 does not directly indicate what such an event would look like, although it does suggest hyper-inflation. In economic accounting, the GDP deflator would be sufficiently large for the inflation-adjusted real GDP to be much lower than the nominal GDP. Historically, hyper-inflation has been associated with periods of societal contraction (Zhang et al., 2007) suggesting some link between current economic inflation and the fraying of previously built societal networks (Garrett, 2012).

On the topic of climate policy, the constant value for $w$ described here does not forbid economic production to become decoupled from carbon dioxide emissions. However, the switch from carbon fuels to renewables or nuclear energy would need to be extraordinarily rapid. Simply to stabilize carbon emissions, much less reduce them, any newly added energy production must be carbon emissions free. Based on recent consumption growth rates, this works out to about 1 Gigawatt of non-carbon energy per day. Alternatively, or concurrently, some means would need to be devised for decoupling historically cumulative wealth $W$ from current energy consumption $E$, effectively by increasing the value of $w = W/E$. Given the value of $w$ has varied so little over the last 50 years, a period during which society changed tremendously, it is difficult to conceive how this would be managed. That said, adjusting $w$ upward could be seen as a new target for mitigating future climate damages.

## Appendix A: Methods

Yearly statistics for world primary energy $E_i$ are available for both consumption and production from the Energy Information Administration (EIA) of the U.S. Department of Energy (DOE) for the period 1980 through 2018, and for consumption from British Petroleum (BP) for the years 1965 through 2019 (DOE, 2020; Bri, 2020). A yearly composite of $E_i$ in units of Exajoules per year for the years 1970 to 2019 is created from the average of the three datasets while using single sources where only one is available. The difference between the values in the BP and EIA data sets is significant, $8.5 \pm 1.5\%$, but it is steady, and small relative to the 180% increase in energy consumption over the 50-year time period considered here. Economic production is tallied and averaged using World Bank (WB) and United Nations (UN) statistics for the years 1970 to 2019 (The World Bank, 2019; UNs, 2020) and expressed here in units of trillions of market exchange rate, inflation-adjusted "real" year 2019 dollars. Statistics for the aggregated capital stock of 180 countries $K_i$ are available from the Penn World Tables (PWT) (Feenstra et al., 2015). Uncertainties in UN, WB and PWT economic values are not published. They are assumed here, as with the energy estimates, to be small compared to the many factor increase in their sizes.

The world historically cumulative production $W_i = \sum_{j=1}^{i} Y_j$ requires for its calculation yearly estimates of $Y_j$ prior to 1970, for which we apply a cubic spline fit to the Maddison Database (Maddison, 2003) for years after 1 C.E. The dataset is adjusted for inflation and to convert from currency expressed in purchasing power parity dollars to market exchange units using as a basis for adjustment the time period between 1970 and 1992 for which concurrent MER and PPP statistics are available. The value for cumulative production in 1 C.E. $W(1)$ is obtained by assuming that $W$ was growing as fast as population at that time at rate $R_W = d \ln W / dt$ and that $Y(1) = R_W W(1)$. Population data from 1.C.E and one century before and after suggest that global population was 170 million and growing at 0.059 % per year (United States Census Bureau, 2021). While there are inevitable uncertainties in the reconstruction of $W$ as with any other, the yearly values of $W$ since 1970 that are emphasized here cover two-thirds of total growth and so the calculations are more strongly weighted by recent data that is presumably most accurate. Thus, calculation of $W$, most particularly the conclusion that $w$ is nearly a constant, can be shown to be relatively insensitive to uncertainty in the older statistics (Garrett et al., 2020).

*Author contributions.* T.J.G. and M.R.G conceived the study, T.J.G. analysed the results. All authors wrote and reviewed the manuscript.

*Competing interests.* The authors declare no competing interests.

*Acknowledgements.* This work was supported by the National Institute of Economic and Social Research and the Economic and Social Research Council (ES/R00787X/1), whose views it does not represent. Review comments from Peter Haff contributed substantially to framing of the arguments.

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
