# Peer review of "Lotka's Wheel and the long arm of history: how does the distant past determine today's global rate of energy consumption?"

_Earth System Dynamics, 2021_

## Author Comment (AC1)

**Response to Richard Rosen**

We place the reviewer comments in bold font and our response in normal font.

**It has been long known that the ratio of energy consumption to annual GDP has been falling at somewhere between 1-2% per year, depending on the year. This implies that energy consumption will slowly fall with respect to cumulative production as well. But this is merely a matter of math and not a cause and effect relationship, since the technologies used for production many decades ago can not affect energy consumption today, except to the extent that a few of such technologies still consume energy. Whatever the lifetime is for old energy consuming technologies, this fact would say little about how fast energy consumption could be made to drop each year in the future. With strong energy efficiency policies in place, energy usage could be made to drop much faster in the future than it has averaged in the past. For example, all electric vehicles which are good for mitigating climate change are far most energy efficient than the currently fleet of vehicles. All electric vehicles could be phased in within 20 years. Similarly, old buildings could be rapidly renovated to reduce their energy consumption. The authors demonstrate that there is a lot of "momentum" built into the energy/economic system, with a fairly constant "velocity" in the past. With enough policy "force" applied to the system, this velocity could be greatly slowed down, as we all hope will happen as climate change is rapidly mitigated, to use a Newtonian metaphor! Thus, the world is not constrained by past energy consumption trends.**

As stated in the paper, the ratio of energy consumption to annual GDP has been falling at a rate of 1.00% per year (line 46). There is no mention of "momentum" or "force" in the paper, as it is not clear that these are well-suited concepts for description of an open thermodynamic system such as the global economy. That said, it follows from Equation 3 that increasing energy productivity $Y/E$ leads to increased energy consumption. Namely, dividing Equation 3 by $E$ leads to

$$\frac{Y}{E} = w \frac{1}{E} \frac{dE}{dt} \tag{1}$$

where $w = W/E$ was found to be nearly a constant for the 50 year period of growth considered. If $Y/E$ increases, due for example more efficient vehicles or buildings, energy demands would also increase. Production grows the system, in which case energy consumption must adjust to sustain this addition in civilization size, as quantifiable through the link to historically cumulative past production given by Equation 2.

**Again, as I indicate in my other comment, the culprit that determines all the energy consumption trends is the type of technologies invested in each year over the past, which consumes somewhat varying amounts of energy from year to year, but which has a finite lifetime. Given the typical rate of global growth over the past decades, the typical energy consuming technology might only be 10-20 years into a 50 year lifetime, to use rough but illustrative numbers. This could be a piece of industrial equipment, a power plant, or a building heating system. Energy consuming vehicles tend to turn over at a faster rate, of course, than once every 50 years. That implies that typically if no new policies are introduced by governments to phase out existing energy consuming technologies, or if market forces do not lead to existing energy consuming technologies being abandoned prior to their normal lifetime, the well-known slow but steady 1-2 percent per year declining trend of total energy use in a given year per dollar of GDP will continue unabated. This is the rough trend that these authors show, however it is precisely expressed. Overall, macroeconomic production functions have little to do with these consumption trends for energy technologies. This argument applies to both fossil fuel consuming technologies as well as to renewable energy consuming technologies. The fairly steady ratio that the authors find between annual energy consumption and cumulative GDP is mostly a coincidence, and a simple product of these slow long term trends for the very slow turnover of energy consuming technologies. Obviously, in terms of cause and effect, only the fraction of any year's GDP that is directly invested in energy consuming technologies cause a fraction of future year's energy consumption until that piece of technology is retired. Thus, macroeconomic arguments alone can never explain these trends. Once either the governments of the world or market forces cause more efficient energy**

**consuming technologies to be invested in more rapidly than typically happened in the past, then this fairly constant ratio can change. The author's analysis shows that this has not yet happened in the past. But if this article is to be published, it must be completely revised so that it focuses on the types and rates of investment in energy consuming technologies**
45 **in each year in the past compared to total GDP in each year. This will allow the authors to explain the trends they find by disaggregating the causes and effects of the trend in terms of technology and not abstract arguments.**

The relationship given by Equation 2 covers a period during which energy consumption $E$ increased by nearly a factor of 3, and economic production by over a factor of 4. However the term "coincidence" is meant to be used in the comment above, a great deal happened in society over this 50-year time period, including a 60% increase in production efficiency, during which
50 the ratio between energy consumption $E$ and historically cumulative production $W$ did not change.

It is difficult to see how to meaningfully disaggregate components of the global system for global scale questions, those most relevant to e.g. discussions of anthropogenic carbon dioxide emissions. No person, machine, or economic sector can be meaningfully disconnected from the global economy as a whole, or even from any time in history, as all are connected through either current networks or temporal causality. The reviewer argues that some fraction of the GDP going towards "energy
55 consuming technologies" while (presumably) the remainder of the GDP goes to production of goods and services that do not require energy. Thermodynamically speaking, this argument is peculiar as nothing can happen production-wise, of whatever sort, without energy having been consumed to convert raw materials into the makeup of civilization, or to make something move faster. And once civilization is grown through production $Y$, more energy $E$ is required to sustain circulations within the previously accumulated internal civilization networks connecting people and their machines $W$. This is even more the case
60 if the production was done efficiently leading to faster growth in $W$. If there is an alternative conclusion that can be drawn from the data presented in Figure 1, it should be presented through consideration of the human system as a whole not by disaggregation into its parts.

---

## Author Comment (AC2)

**Response to Reviewer 1**

We place the reviewer comments in bold font and our response in normal font.

**(1). I object to arbitrarily introducing a new production function (equation 3) without serious discussion. The discussion in the text, based on curves in Figure 1 is not nearly sufficient to justify equation 3. The standard Cobb-Douglas production function was introduced in 1928 for a good reason and the other production functions economists have introduced and tested since 1928, have histories also. I am not defending any of them, but the reasoning behind equation 3 cannot simply be based on the data represented by the curves in Figure 1.**

We agree that Equation 3 is not a production function in the same sense as e.g. the Cobb-Douglas or Leontieff production functions, namely a function that calculates the output of economic production $Y$ on the basis of inputs, or factors of production, typically capital $K$ and labour $L$. As mentioned in the first paragraph, such "production functions" (deliberately put in quotation marks on line 16), grossly underestimate the role of energy in production by treating the energy sector as just one of many other sectors in the economy. By contrast, we are interested in the direct relationship between primary energy consumption and economic output, without going through the route of a production function in the economic sense. For this reason, we termed Equation 3 an "inflation-adjusted production relation" on line 67 (immediately before Equation 3), but regrettably used the term "production function" on line 72 (immediately after Equation 3), which we will change to "production relation".

Having said that, the result expressed in Equation 3 is a direct consequence of the 50 years of data in Figure 1 supporting a nearly fixed relationship between historically cumulative GDP and current energy consumption as given by $W(t) = \int_0^t Y(t')dt' = wE(t)$. Namely, we first *define* $W$ as the integral of production $Y$ in Equation 1 and then *observe* in Equation 2 that this quantity is related to energy $E$ through a nearly constant ratio $w$ over a period during which the GDP increased by a factor of 4.5. Equation 3 then follows purely mathematically from Equations 1 and 2 (namely by taking derivatives and assuming that $w$ is constant), that is, $Y(t) = wdE/dt$. Equation 4 then further validates this relation by estimating $w$ as the ratio between $Y$ and $dE/dt$, with a value similar to the one found in Equation 2.

The physical arguments supporting why might the relationship hold are as described by Lotka (1922) in the introduction and elaborated upon in the discussion. It is not clear from the comment why empirical evidence and physical reasoning is considered "arbitrary" as the basis for establishing a link between economic production and growth in energy consumption.

**(2). In another place, the authors note that the usual relationship for capital growth is [dK/dt = Y-C minus delta (capital depreciation)], where delta (the depreciation rate) is constant and C = cY where c also is assumed to be a constant. The last assumption is wrong. The ratio c = C/Y (the fraction consumed) may leave a significant surplus for capital investment now (and for the past 50 years) but 200+ years ago c was practically unity while the depreciation rate was smaller than it is now – the surplus for investment or saving back then was virtually zero, and what surplus there was came from coal mines. In other words, until very recently almost everybody needed every bit of their income to buy consumables, mainly food and fuel (for light and heat). So, in the long run c is not a constant; it can (and will) decrease. Neither is the depreciation rate constant, by the way. Most people will spend their time playing computer games.**

Nowhere in the article is it stated as the reviewer claims that depreciation or the ratio of consumption to production $c$ is constant. See lines 100 to 105 which in fact point to the opposite. Average rates are provided, but this is stated by acknowledging that the instantaneous rates have changed during the period under consideration. The point being made is that whatever new that is produced is either consumed or depreciates at a rate that appears too rapid to allow for societal contributions from the more distant past.

**Also (3) the curve shown for capital stock K in Figure 1 is presumably based on prior work by Garrett but – being central for the rest of the argument – the underlying data also needs explanation and justification, especially since**

**Garret's earlier work in this field has not been widely accepted. (That is not a criticism). The underlying capital stock K data for Figure 1 should be published.**

The underlying capital stock data are from the Penn World Tables and are referenced, as are all other quantities. See the Appendix for methods.

---

## Author Comment (AC4)

**Response to Peter Haff**

We place the reviewer comments in bold font and our response in normal font.

**The arguments of this paper represent a conceptual break with standard views of the role of energy in modern society. As such, its claims are likely to be controversial. However, if the thesis of this paper holds up, it would stand as a significant advance in our understanding of the limits to human control of energy consumption by the global human-technology system (abbreviated in this review as the "world system"). In their explanations in the paper, and in author commentary on this site responding to reports of other reviewers, the authors' have defended their methods and conclusions to my satisfaction. I recommend publication of the paper following the authors' response to specific points and recommendations made at the end of this report.**

We sincerely appreciate the supportive remarks.

**The authors' conclusions, based on their analysis of economic time series, imply that global energy consumption cannot be manipulated at will, at least not without radical disturbance of the modern world system. This should not be unexpected since energy is not like other commodities, but plays a special role in any dynamic system. Thus, for the world system, the energy slice of the economic pie, perhaps totaling to 10%, is unique, in that it pervades and powers every other sector. Removing half of a 10% energy sector would not be a 5% overall economic effect, but more like a 50% effect, or larger, somewhat like the consequence for a person of the removal of one half of their rather small volume of blood. The authors report the counterintuitive result that the current rate of energy consumption by the world system is proportional to world economic production ("GDP") summed over all past years. This implies that current energy consumption is driven not just by recent economic activity, where human influence on energy use is most obvious, but also by activity performed in the distant past, whose consequences remain with us today.**

We agree with the reviewer's summary of the article.

**One obvious criticism of this claim of historical determinism is the observation that physical devices and systems of long past years that no longer function, or that no longer even exist, cannot continue to consume energy today and thus cannot contribute to current energy use. The authors' explanation of the enduring influence of past production on current energy consumption is that individual system components existing today, together with their potential energy demands, are generated not just through recent production but also in consequence of many years of prior production. This seems to me a reasonable argument. Thus, the magnitude of current energy use (i.e., the system's metabolic rate) depends on the total value of production achieved along the specific trajectory of the system's historical development, because that production necessarily generated incremental additions to past metabolic base rate that made possible the existence and performance of those populations, cultures, plans, patterns of organization, tools, inventions, wars, and other factors that did in fact elevate consumption over time to the level seen today. That is, construction of the system, a result of chronic non-zero GDP, represents more than accretion and organization of material into a complex growing edifice, but also requires a continuous and dedicated flow of energy to support what has been created. Thus, a suitable and growing metabolic base rate E has to be maintained at all times in the system's history. In the simplest model the background energy demand that sustains this construct—the world system–would simply be proportional to the total production. Although old material construction disappears with time and technology becomes more efficient, complexity and size of the system continue to increase, suggesting that energy demand of these latter factors outweigh what is lost through decay and increased efficiency of specific devices and systems.**

The wording employed here is superb. We hope to include a modified form of the phrasing in a revised manuscript.

40    **Another general critique that might be raised is that the authors emphasize physical principles in their analysis of large-scale societal energy-use rather than turning to more standard tools of economics. However, an approach informed by physical requirements that are applicable to any dynamic system may be more suitable for broad, global analyses where smaller-scale details fade away, than methods based on economic assumptions and models calibrated to influences of national or local markets and to cultural behaviors. Thus, for example, the law of energy conservation, the**
45    **2nd law of thermodynamics and the requirements of dimensional consistency in equations and variables reveal their utility when applied at global scale. Models that employ non-scalable relations or manipulate quantities that cannot be clearly connected to basic physical variables may be useful tools for specific applications where careful calibration is possible, but these approaches will generally be less useful for questions that require extrapolation outside the restricted problem-space for which they were designed.**

50    We agree. It works both ways. The simplicity of the approach presented here makes it less well adapted, at least a priori, for studies of economic sectors or nations. The reverse is also likely to be true, that traditional economic models are less able to address world-scale economic behaviors that are constrained by such key issues as resource availability.

    **The conclusions of the paper, if they stand up to future criticism, have substantial implications for future human well-being. They point to a fundamental challenge facing efforts to manage world energy consumption. The proportionality**
55    **of current energy use to total past production suggests the difficulty of redesigning a system when the past holds sway, a point illustrated in microcosm by the expense, disruption and political resistance that often accompanies attempts to renew or replace urban infrastructure in long established cities. Change, by contrast, can be much easier to effect when it occurs as growth, i.e., a positive GDP—in the above example perhaps outward expansion of the city—rather than as a reconstruction of legacy systems which, through lack of access to other sources of energy, are forced to disrupt and**
60    **cannibalize the extant metabolic energy flows that sustain their existence.**

    **Two main points I take away from the paper are: 1. That the influence of the past infuses an intrinsic conservatism into the dynamics of the world system, according to which it tends to resist change; and 2. In such a physically enforced conservative environment, significant change and the production of novelty are made possible only by injecting energy into the world system faster than it is dissipated by its underlying metabolic processes, i.e., by increasing the rate of**
65    **world energy consumption and in the process adding to total past consumption.**

    This is well worded.

    **Title: I would change the title of the paper to something more compelling. Many in the ESD community who might be interested in its arguments may pass over what may appear as a discussion of the minutiae of economic production. For example: "Lotka's wheel and the long arm of history: how old technology and forgotten ideas determine the**
70    **value of today's global rate of energy consumption", or, perhaps: " 'The past isn't dead. It's not even past': how old technology…". (Faulkner).**

    These are excellent suggestions. We adopted the following modified version of the first one: "Lotka's Wheel and the long arm of history: how does the distant past determine today's global rate of energy consumption?".

    **I might also suggest adding a sentence or two emphasizing (as per the urban infrastructure example above) that**
75    **change and novelty ride on the back of dE/dt, not E, the latter of which supports business as usual.**

    We have added the text: *that is the innovation of something newly consumptive over and beyond the business-as-usual scenario that would have energy consumption rates stay constant.*

    **Line 30: "wit units". There are a few typos in the text. These should be removed using a fine tooth comb; they indicate momentary lapses of attention, a condition which leads to doubts in readers' minds about bigger issues**

80    We have carefully reviewed the text.

**Line 31: The paper states that it focuses only on global quantities, but use of "Gross Domestic Product" or "GDP" with no qualifier may cause confusion. Perhaps settle on uniform use of "Gross World Product" or "global Gross Domestic Product" instead?**

We have added the qualifier "world" in several places prior to GDP. GWP turns out to have its own problems due to the
85   potential confusion with Global Warming Potential.

**Line 57: Clarify explanation of why the relation between W and E is not simply one of correlation.**

The text has now been rewritten to better explain the intended points.

*The relationship between $W$ and $E$ does not appear to be one only of correlation between two quantities, as for example has been noted for $E$ and $Y$ (Jarvis, 2018). Instead the two quantities have maintained a linear scaling over the half century*
90   *period for which widely published data are available. A least-squares fit to the logarithms of $W$ and $E$ yields the relationship $W = 5.47E^{1.00}$ Calculated instead as a linear fit, the relevant expression is $W = 5.67E - 66$. Note the intercept of the fit where $E = 0$ is equivalent to $W = -66$ trillion 2019 USD. This value is just -1.9% of the 2019 value for $W_i$ of 3547 trillion 2019 USD, and so sufficiently small as to plausibly approximate the origin. By contrast, the linear fit for world GDP and energy is $Y = 0.17E - 21$. So, while $Y$ and $E$ are correlated, they do not scale since the intercept corresponding with zero energy*
95   *demands is $Y = -21$ trillion 2019 USD, or -25% of the 2019 value.*

**Line 73: "non-integer exponent of E". Perhaps expand discussion here slightly to illustrate the problem of scaling with variables or exponents that have been determined simply by familiarity and/or calibration rather than by relation to actual physical process. This can also help emphasize the value, where it is appropriate, of a physical framework in place of traditional, non-physical models. In general this paper is a good venue to expand the argument for treating**
100  **(some) economic problems in a framework that manifestly respects, or at least does not contradict, physical law.**

The text has been revised to read

*Expressing economic production as related to a change in energy demands, that is its derivative with respect to time, differs significantly from prior approaches. These most usually employ production functions that ignore the role of energy altogether. In the few studies where production functions do appeal to energetic demands, the functional dependence is to some non-unity*
105  *exponent of $E$ (Ayres et al., 2003; Keen et al., 2019), which physically and dimensionally is nonsensical. While it is certainly possible to obtain through fitting a non-unity exponent relating two quantities, it cannot be presumed the fit expresses something fundamental about the system unless the appropriate units for physical quantities are maintained.*

**Line 127: "provided the system is in its phase of growth". This caveat may not be needed. Thus, using a biological example, mature organisms whose growth has stopped nonetheless still consume energy at a steady rate, E, determined**
110  **by their size (total production), but their "GDP" is zero. For such systems where E is nominally a constant (no growth), minor wear and tear might be fixable by the system, but accumulating insults to functionality with age, or more catastrophic impacts which the system is ill equipped to combat using already spoken for metabolic energy supplies, may eventually make it impossible for the system to survive (maintain its metabolic rate) in the absence of access to increased sources of external energy. The constraint of a constant metabolic rate makes organismal longevity a challenge, and in**
115  **the end a losing battle (as in the case of an organism). The results of the present paper suggest that a continuing increase in energy consumption is a necessary condition for the long term survival of world civilization. Of course it is not a sufficient condition. A state of chronic acceleration cannot last forever and limiting effects that are outside the scope of this paper will eventually have their impact.**

Point taken. The text has been reworded slightly to read:

120 *But, in all cases past consumption is a primary determinant of the system's current energetic demands.*

**Finally, the other reviewers raise interesting points that together with the responses of the author help to clarify the arguments of the present paper. I believe the authors can further improve their paper by incorporating into it some portion of their written responses to these reviewer/commenter suggestions. I hope my own review is similarly serviceable, and that, after manuscript revision, which I believe does not need to involve a major rewrite, ESD will**
125 **proceed to publication of the manuscript.**

Our sincere appreciation for a thoughtful review.

**References**

Ayres, R. U., Ayres, L. W., and Warr, B.: Exergy, power and work in the US economy, 1900-1998, Energy, 28, 219–273, https://doi.org/10.1016/S0360-5442(02)00089-0, 2003.

130    Jarvis, A.: Energy Returns and The Long-run Growth of Global Industrial Society, Ecological Economics, 146, 722 – 729, https://doi.org/https://doi.org/10.1016/j.ecolecon.2017.11.005, 2018.

Keen, S., Ayres, R. U., and Standish, R.: A Note on the Role of Energy in Production, Ecological Economics, 157, 40–46, 2019.

---

## Referee Report (RR1)

The authors have clarified their arguments and substantially addressed major referee concerns. I recommend publication of the ms after some attention to the following points. Namely, there are a couple of small changes that the authors could still make that I think would go even further in aiding the reader.

The variable $E$ is defined (line 32) as the instantaneous rate of world energy consumption. $E$ is power, but looks like it should be energy. So I would suggest that variables that are rates rather than stocks/quantities should be designated by an overdot or other indicator, thus $E \rightarrow \dot{E}$ (energy/time), where $E$ now is indeed energy. This is trivial, but such changes will make the ms more accessible, especially to the "casual" reader. Similar changes might be made for (instantaneous) world GDP, $Y \rightarrow \dot{Y}$ (USD/time), and for $K$. Fig 1. would remain (almost) correct because $E$ as plotted would indeed be energy (although not instantaneous energy, but rather $E_i$, as already explained in the caption, and similarly for the other variables there). In Eq (2), because $W$ as well as $E$ (i.e., $\dot{E}$) are instantaneous values, the units of the constant $w$ would be essentially USD/watt (not USD/energy/year).

Most importantly, the dot notation, if used in Eq (3), $\dot{Y} = w \dfrac{d\dot{E}}{dt}$, makes clear(er), as pointed out in the Conclusions (line 168), that even if GDP ($\dot{Y}$) is not increasing, as long as it is positive, energy consumption accelerates. So a fixed value of GDP, even if small, means that demands on energy (and thus material) resources will not remain fixed, but will continue to increase.

Eq. (3) is in my view a key summary of the results of the present paper. As such, its implications might be emphasized a little more fully. Thus, decreasing GDP to a small positive value only means that any problems associated with increasing use of energy will continue getting worse, just not as fast as before. However, if GDP actually goes to zero (no economic growth), then energy use does not accelerate, $\dot{E} = const$. For some, this no-growth condition may approximate an imagined ideal of a "steady-state" world. But in this case $\dot{E}$ constitutes the base metabolic rate of the world economy, a level of energy consumption too small to deal effectively with large-scale environmental disruptions of society or for developing and deploying new tools for managing/improving human health (e.g., cancer) or social infrastructure (e.g., housing, basic income). Re environmental challenges, one might consider the case of an adult animal, which is a no-growth system. It has enough energy reserves to accelerate (by running away) to escape a predator, but not enough to outrun a larger and more sustained threat, like a forest fire. Regarding civilization, because the occurrence of future global emergencies is virtually guaranteed, especially in an ecologically distressed world, a condition of no-growth, $\dot{E} = const$, would not be hospitable, or perhaps even survivable, for the collective human enterprise no matter how sophisticated the civilization. A condition of degrowth, i.e., $\dot{E} < 0$, would be a recipe for collapse of the technosphere (global civilization), a result stated in reverse by the authors, namely that collapse is a recipe, perhaps the only recipe, for degrowth.

The final comment of the ms, line 180, on managing the constant $w$, is interesting. It points to the unknown details of how exactly past production retains its currency in the modern world even after many years of physical decay. This is a topic, among others suggested by the paper, that deserves more

study in the future, and as such is a clear sign of the value of the present work not only for its analysis but as a stimulus for further research.

Finally, the conclusions of the paper are of course provisional and might be in error. Nonetheless, publication will likely generate substantial controversy and pushback because of its message that there are hard-to-change, or even potentially unchangeable, structural obstacles to managing the human future in ways that are palatable to many potential readers. Generating friction is an occupational hazard for those who try to dispassionately separate the workings of the world as implied by scientific analysis from a vision of how one might wish the world to be. Where friction exists, and at the same time scientific analysis seems as sound as it can be given the uncertainties of the problem, wide dissemination of the results can be especially valuable.

---

## Referee Report (RR2)

The current revision of this manuscript clarifies earlier arguments of the authors Garrett et al and adds support to my previous recommendation for proceeding with publication. I summarize their main argument and revisit its implications in order to reinforce that recommendation.

The key result is the claim that the future behavior of the human-technological "world-system" (civilization, global society, technosphere…) is strongly constrained by a single quantity, W, a measure of the accumulated products of its past consumption of energy. The reason that significant aspects of the behavior of something as complex as the world-system can be subjugated to a seemingly simple quantity like W is that complex systems are directly describable at a given scale (here the global scale) only by concepts and constructs native to that scale. Considered in its relation to global consumption of energy, W appears to be such a construct.

Garrett et al appeal to economic data to argue that over time energy consumption E by the world-system is proportional to W. This relationship has the alarming and perhaps puzzling consequence that the rate of energy consumption by the world-system today seems to be wired into its past history of production through a simple dependence on W. The authors illustrate their resolution of the puzzle through various analogies showing how energy use by diverse systems (for example a tree) reflects their cumulative productive history, even if past production does not remain metabolically active (e.g., the dead heartwood of the tree). The significant point is not just that consequences of prior human behavior, in the form of past production, still influence human actions today, since of course the present is contingent on the past, but rather that the rate at which society uses energy seems to be determined by the value of W. Humanity appears destined to experience the consequences of continuing increases in energy use given that acceleration is the source of new productivity which in turn adds to the value of W. If all human endeavor, as far as energy use today is concerned, is indeed wrapped up in the quantity W, the latitude for human action to shape the future of civilization would seem to be severely limited.

The Garrett et al argument is formulated within a physical framework that focuses on the underlying condition that drives essentially every global Anthropocene crisis from climate change to occurrence of pandemics to spread of mass surveillance, namely, the chronic acceleration of energy use. The ideas and connections discussed are interesting, provocative, and, in my view, well-defended, and they open many new avenues for thought and research. The authors' conclusions should be more widely known, not because the issues are settled, but to open them to wider consideration and debate.

In preparing this review I composed several more pages of commentary. However, on reflection that seemed to be overkill, so rather than continuing I stop here. As part of the review process, however, I would be happy to respond to any remarks or questions by the ESD community.

---

## Author Response (AR2)

Minor word-smithing was done throughout the document for style and typos.

**Response to Referee 1**

We thank once again Peter Haff for his insightful, constructive comments. We place comments in bold font and our response in normal font.

5 **The variable E is defined (line 32) as the instantaneous rate of world energy consumption. E is power, but looks like it should be energy. So I would suggest that variables that are rates rather than stocks/quantities should be designated by an overdot or other indicator...**

This is difficult. At one level, we entirely agree. At another, we have decided with some misgivings to adopt the traditional notation used in energy economics where $E$ and $Y$ are both rates. For example, if we used $\dot{Y}$, which is common in economics,
10 it would be almost certainly misinterpreted as a term expressing economic growth, precisely what we want to avoid. What we have done to address the concern is clarify that $E_i$ and $Y_i$ remain rates, effectively yearly averages of the instantaneous quantities $E$ and $Y$.

**Eq. (3) is in my view a key summary of the results of the present paper. As such, its implications might be emphasized a little more fully...**

15 We have added the following paragraph:

*In the hypothetical limiting case of $dE/dt = 0$, the world attains a sort of metabolic steady-state characterized by a balance between energetic and material inputs and outputs. Energy consumption maintains a fixed rate, but also there is no real economic production. Nominal production may remain, but it is completely eroded by inflation. The case of economic collapse may not be survivable. If so, the point at which $dE/dt = 0$ may only represent a temporary marker on a pathway to more*
20 *complete thermodynamic collapse with the steady-state condition of $E = 0$. A distinction must be made with the quite different steady-state condition where $dY/dt = 0$, namely one of constant inflation-adjusted economic production $Y$, or zero GDP growth, as this would imply continued expansion of energy demands at rate $Y/w$. In the constant GDP growth case with fixed $d\ln Y/dt$ energy consumption accelerates.*

**Response to Referee 2**

25 We place comments in bold font and our response in normal font.

**Garrett's current paper appears to use roughly the same methods as his 2009 Climatic Change article, which Steve Schneider asked Danny Cullenward and Lee Schipper to review. Garrett doesn't cite this critique or otherwise discuss its concerns. Nor does he address the critical differences between exergy and the primary energy consumption accounting paradigm reported in the BP statistics. He just runs some new data through the same basic calculations.**
30 **The Cullenward et al response lays out the problems with Garrett's work very clearly, and it applies just as well to Garrett's latest article. The abstract summarizes things well:**

**"Uncertainty in the trajectories of the global energy and economic systems vexes the climate science community. While it is tempting to reduce uncertainty by searching for deterministic rules governing the link between energy consumption and economic output, this article discusses some of the problems that follow from such an approach. We**
35 **argue that the theoretical and empirical evidence supports the view that energy and economic systems are dynamic, and unlikely to be predictable via the application of simple rules. Encouraging more research seeking to reduce uncertainty in forecasting would likely be valuable, but any results should reflect the tentative and exploratory nature of the subject matter."**

According to the editor Michael Oppenheimer handling the Cullenward article, it was not peer-reviewed. The article included
40 personal attacks by way of reference to a cartoon and no quantitative analysis to support the contention in the abstract repeated above. Further, the article claimed that the scaling between energy and cumulative production we claimed in Climatic Change, and through more extensive analysis in the ESDD article here, was incorrect by making a straw man argument: there does not exist a fixed ratio between energy consumption and economic output. Specifically, Cullenward et al state " Indeed, the trend we observe in the data—a declining energy/GDP ratio—would seem to be the counterargument to Garrett's model formulation, per

45     inflation-adjusted 1990 USD.". This changing E/Y ratio, which is true, was explicitly noted both in the Climatic Change paper and in the article here. Figure 1 of this paper show a declining energy/GDP ratio, precisely what the reviewer and Cullenward claim invalidates our argument. It appears that neither Cullenward et al., nor the reviewer, understand the primary thrust of the arguments we make, a basic relationship taught in introductory calculus, or even arrived from casual observation, that cumulative and instantaneous quantities do not evolve at the same rates.

50     **In terms of specific issues with Garrett's analysis, there are a few key points:**

    **1) Primary energy is calculated in different ways, and it's important to correct for differences between different sources. Garrett's averaging of primary energy statistics between sources that use inconsistent conventions for estimating primary energy of non-combustion resources (eg EIA and BP) shows that he doesn't appear to understand these subtleties.**

55     Taking the period from 1980 to 2018 for which both EIA and BP values are published, the difference between the two datasets is small relative to the growth, that is the difference is $8.5 \pm 1.5\%$ compared to a near energy consumption tripling. There is no trend in the difference either. So, whatever the subtleties in the choices made by the EIA and BP, the difference does not affect the constant scaling claimed in the article. We have added to the appendix:

    *The difference between the values in the BP and EIA data sets is significant, $8.5 \pm 1.5\%$, but it is steady, and small relative*
60 *to the 180% increase in energy consumption over the time period considered here.*

    **2) Primary energy is the wrong metric. Exergy accounts for both primary energy and ENERGY QUALITY (which reflects different sources' ability to do work),and that's what any analysis of long-term trends should be using. Electricity has the highest exergy of all energy forms, and the substantial shift in the percentage contributions of electricity vs. other fuels over time shows that this effect can't be ignored.**

65     There appears to be some confusion in this remark about the term exergy and primary energy. Exergy is jargon for the amount of energy that is available to do work. Primary energy is also a form of exergy, as it too represents the potential energy for work by civilization as a whole. Now, in energy economics, a more restrictive definition of exergy is often employed wherein, as the reviewer points out, electricity has higher exergy than, say, combustion. Electricity in a toaster makes toast and heats the home, whereas natural gas only heats. This distinction may be useful from the perspective of a home, but doesn't account for
70 the energy that must have been made available to produce electricity. Ultimately, all potential energy that is available to enable civilization activities of whatever kind must be traced back to primary energy. Or, all primary energy consumed by civilization is used to sustain civilization activities, however inefficiently, ending its lifetime as waste heat, which is eventually radiated to space. Now, there is a slight subtlety, which is that a small fraction of this energy is available to contribute to the growth of civilization. The characteristic timescale for sustenance of civilization activities is most probably one day. The characteristic
75 time scale for growth, taking civilization's energy consumption growth rate to be 2.2% per year, is about 30 years. Thus, the two basic processes of sustenance and growth through consumption of civilization "exergy" can be readily separated. We claim that the energy/exergy required to sustain civilization's daily activities, the primary energy, is tied to historically cumulative real economic production. That is what is supported by the data.

    The text beginning Section 2 now reads

80     *To avoid complications associated with the details of trade, interactions between economic sectors, or distinctions between energy types, this study is focused only on global quantities, as described in the Materials and Methods below. Annual primary energy sources, those that are available to drive civilization activities of whatever type, are consumed and ultimately dissipated as waste heat at a rate that can be expressed as an instantaneous quantity $E$ (e.g., Terawatts) or a yearly-averaged quantity $E_i$ with units of power (e.g., either Terawatts or Exajoules per year) (Garrett et al., 2020).*

85     **3) GDP is a terrible metric of human activity and well being. In addition, changes in the structure of the economy over time over decades are so large as to make the expectation of structural constancy unreasonable.**

    The paper is not focused on discussions of metrics of human activity or well-being, which we would agree are not well characterized by the GDP. GDP is purely a financial quantity that we show can be linked to a thermodynamic quantity. Civilization activities encompass far more than human concerns, including flows along networks that include machines, mines,
90 roads, shipping routes and much more. We are unsure what structural constancy is being referred to or, how our study neglects it as the article explicitly remarks on how much has changed in 50 years at the beginning of Section 3. We only assume that thermodynamic laws are constant over the time period being considered.

**4) Treating energy and economic statistics like they are as reliable as measurements of physical systems is a mistake that many physicists make, but it's a category error. Physical determinacy for economic systems is a logical fallacy to which some in the degrowth community (among others) appear to be susceptible, but it's not a justifiable intellectual position.**

All physical measurements are uncertain, and to widely varying degrees. Witness for example the burgeoning study of exoplanetary properties, where almost nothing is known. Unfortunately economic statistics are often not reported with error bars. For example, world GDP is reported by the World Bank and United Nations to either the dime or the penny. While, presumably, these statistics are far more uncertain than such precision, no readily available basis is provided for estimating the magnitude of the uncertainty. That said, the results presented here are relatively insensitive to relative uncertainty in yearly economic statistics because a 50-year time series is considered over which GDP increased by a factor of 4.5.

The text in the Appendix now reads:

*Uncertainties in UN, WB and PWT economic values are not published. They are assumed here, as with the energy estimates, to be small compared to the many factor increase in their sizes.*

**Response to Referee 3**

We thank the reviewer for constructive comments and place the reviewer comments in bold font and our response in normal font.

**I have read the paper and the other reviews. I have a couple of small criticisms that can be fixed quite easily and a couple of others that might need more thought. The first criticism (p 4, is that their comment about production functions incorporating E explicitly is incomplete and inaccurate. The LINEX function used by Ayres & Warr, where E is a third factor and the variant where E is replaced by useful work U are explained in detail in chapters 5 and 6 with numerical results in Chapter 7 of "The Economic Growth Engine'' by Ayres and Warr (Elgar, 2009). That reference should be cited in addition to the 2003 paper. Garrett's comment that exponents are "physically and dimensionally non-sensible'' is unjustified and unnecessary. If the comment is important it should be explained better. My second comment is that the discussion of Equation 3 is vary opaque. I've reread it several times and I don't see the point. What I do see is that it could make make sense to express a possible proportionality between capital stock K and total stock W. If not, why not? But that possibility wasn't considered. Wouldn't it make sense? Beyond that, the purpose of the paper is unclear. The discussion of Lotka's wheel is interesting but where does it take us? I am not persuaded that we now have a better production function, especially given the excellent results of the LINEX function and recent developments in that domain.**

The possible proportionality between $K$ and $W$ is shown in Figure 1. They do not scale. Otherwise, the text after Equation 3. has been written to read

*Eq. 3 expressing economic production as proportional to an increase in energy demands, that is its derivative with respect to time, differs from prior approaches that tend to ignore any explicit mention of the role of energy. Where energetic demands are considered, the production functions are complex, and the dimensions of the problem are not considered. Rather than starting with the constraint that the factors of economic production, of whatever combination, must tally dimensionally to units of currency per time, quantities such as dimensionless capital, labor, and useful work are set to non-integer exponents, or are themselves placed in exponents (Ayres et al., 2003; Ayres and Warr, 2009; Lindenberger, 2011; Keen et al., 2019). While the functions can be shown to reproduce past behaviors for specific nations, it is only by way of specifying coefficients, or "output elasticities", that are themselves determined from past economic conditions, and that vary according to the time period considered. The production functions become moving targets, and therefore cannot be presumed to express something fundamental about the economic system. As attributed by E. Fermi to J. von Neumman "with four parameters I can fit an elephant, and with five I can make him wiggle his trunk." Here, by contrast, Eq. 3 is simple, dimensionally reasoned, and assumes that $w$ is a constant, so it can be readily refuted (or supported) with data.*

**Personally, I think more could be done with this material, and it would be interesting to see how it looks just for the US (which used to be quite self-sufficient) back to 1860 or so.**

We appreciate the comment. The remark about self-sufficiency is important however as it would need to be accounted for. For the moment, since the intent is to address such global scale problems as carbon dioxide emissions, we are limiting our studies to global scale quantities.

---

## Author Response (AR3)

**Response to Professor Herrmann-Pillath**

We thank Professor Herrmann-Pillath for the time reading the manuscript and the thought put into the review. Our response is outlined below.

**GDP is often defined as a measure of production, but that is misleading, as is well known. 'Production' means 'eco-**
**nomic value added', hence represents the classical distinction between productive and unproductive activity. If one**
**compares this measure with energy consumption, this mixes entirely different categories, as energy consumption relates**
**to material production (even making Bitcoins is 'material', in the sense of producing certain processes on computers).**
**Material production as a category is much larger than GDP. For GDP, it is necessary to subtract all intermediary**
**production because these are just costs that do not add value. But for considering energy throughput, of course all**
**production matters. That means, I think that the appropriate measure would be gross production, but not GDP. This**
**comes closer to modern approaches of Material Flow Analysis than GDP. Hence, I think the authors should switch**
**to that measure. To avoid misunderstanding: Gross production or output (covered by Input-Output tables in national**
**accounts) is not 'material' in the sense of MFA but is also measured in value terms. The point is that the notion of 'pro-**
**duction', in my view, must include all productive activities, hence also intermediate stages, when exploring regularities**
**in energy flows through the economy. Beyond that, there is much productive activity that is not covered by national**
**accounts, well recognized for long, such as shadow economy and household production. Again, for considering material**
**and energy flows one would need the full picture of economic activity.**

We are aware of the distinction between gross production and the GDP or "value added". However, the article is not attempting to illustrate a relationship between gross production and energy flows, as seems to be suggested by the reviewer: "for considering energy throughput, of course all production matters.". Instead, the focus on "value added" is meant to avoid double-counting, rather than ignore intermediate steps. For example, if the primary cost of extracting wood from a forest to build a shed is $10,000 and this wood is then sold to Home Depot for $12,000, which then sells to my contractor for $15,000, who then charges $20,000 for the shed, the correct accounting for the final product is indeed $20,000, and *not* the sum of all transactions, namely $10,000 + $12,000 + $15,000 + $20,000 = $57,000, as this would be a tremendous overestimate of the original cost of extracting the wood, counted here 4 times. In other words, the "value added" by the timber company is $2,000 (which corresponds to other costs not necessarily the wood itself, such as machinery and labour), that of Home Depot is $3,000 (again, only taking into account previously unaccounted costs such as the staff in the store, etc), and the value added by the contractor is $5,000 (say his labour and equipment).

Thus, the argument being made is that the GDP – precisely as a measure of the addition to economic value – is tied to the *increase* in energy consumption, not to energy consumption itself. This was already stated in Section 3 of the article:

> *Eq. 3 expresses economic production as proportional to an increase in energy demands, that is its derivative with respect to time.*

Regarding the role of e.g. housework, this too is already discussed:

> *By way of explanation, consider the circulations within our bodies, brains, and machines, and our activities such as housework, transport to and from work and the grocery store, and even conversation among family and friends, that all of these require current energy consumption in some form. Each one of these may involve a financial transaction at some prior stage, for cleaning products, gasoline, or food, yet no financially quantifiable purchase is made at the point at which the energy is consumed.*

[Figure]

**Figure 1.** Representation of Lotka's view on the thermodynamic mechanisms governing system growth, involving a wheel that enlarges and accelerates using an "un-utilized residue" of energy and matter

Seeing the connection between past, current, and future consumption is critical. Civilization, as an open thermodynamic system, must consume and dissipate energy to sustain all its activities, insofar as they have developed from the past, whether or not they are tallied in yearly national accounts. Final production, or value added – otherwise termed the GDP – is argued here to depend on an imbalance, whereby consumption exceeds dissipation, allowing civilization to grow to consume more in the future as it becomes embedded in the structure of the network of civilization. The daily imbalance can be quite small, about 0.01% of daily consumption (Garrett et al. 2020), yet still lead to a doubling of thermodynamic demands in about 50 years. Thus, strictly, the inference from a fixed relationships between E and W is that there is a fixed relationship between the GDP and the rate of energy consumption *growth*.

We have attempted throughout the manuscript to further clarify this argument, including an additional quote from Lotka's paper in the introduction:

> *Lotka emphasized that "In every instance considered, natural selection will so operate as to increase the total mass of the organic system, to increase the rate of circulation of matter through the system, and to increase the total energy flux through the system,* so long as there is presented an un-utilized residue of matter and available energy." (our italics).

This is accompanied by a new figure illustrating Lotka's point, Figure 1.

and a figure illustrating our point (Figure 2):

We have also written such statements as:

[Figure]

**Figure 2.** Elaboration on Lotka's Wheel. Civilization growth related to increases in its power at rate $dE/dt$ as tied to network production through the inflation-adjusted GDP $Y$. Current power $E$ is thus tied to the historically cumulative GDP through $W = \int_0^t Y(t)\,dt' = w\int_0^t (dE/dt)\,dt' = wE$

.

> *A consequence of the relationship is that inflation-adjusted economic production is more closely related to a surplus of energy – or the "un-utilized residue" using Lotkas's words – than the rate of energy consumption itself*
>
> *So, it is only with an excess or "un-utilized residue" of available energy that an effective phase change becomes possible whereby raw materials are converted through economic production into newly created civilization networks, and societal movements can be accelerated along them.*

**The other problem is the cumulation of these values. This makes even less sense for GDP than for gross production. The justification forwarded by the authors simply does not apply for value added. If one uses GDP, the only reasonable approach would be capital formation, which could be conceived as 'embodied energy'. For gross production, one might possibly argue that this is somehow becoming embodied in many forms, as the authors try to explain for GDP. But that needs more detailed analysis.**

If the GDP represents "value added" tallied over the (somewhat arbitrarily chosen) course of one year, we are simply using the mathematical property of addition to tally over all of history, making a key downward adjustment for inflation. The interpretation is effectively that the real GDP represents an addition to pre-existing societal networks. But, regardless of

[Figure]

**Figure 3.** The ratio of capital formation values $dK/dt$ to annual energy consumption, setting the ratio in 1970 to 100.

interpretation, or beliefs to the contrary, the data presented here should speak for itself. No criticism has been made by the reviewer (or any other reviewer) of how the analysis was done. As is clear from the figure here, there is no correspondence between energy consumption rates and capital formation as suggested by the reviewer (see Figure 3), only to the historically cumulative world inflation-adjusted GDP as already argued in the article.

We have included the following in the article:

> *A related quantity, the rate of capital formation, $dK/dt$, is not shown because it is implicit in the curve for $K$, however its value varied considerably. While it increased by a factor of 1.5 between 1970 and 2019, the relative increase was 3.2 in 2009 and 0.34 in 1982.*

**That being said, the empirical regularity is worth noting. But I do not think that it can be directly extrapolated as long one uses GDP. I do not defend simplistic decoupling theories, yet it is true that 'value added' has no direct material interpretation but depends on how we measure consumption and production and what specifically is valued on the marketplace. For example, if people start to value playing the flute, they will spend money for buying one flute, perhaps even in their lifetime. Otherwise, they pay for a teacher who does not need any other equipment than her skills and knowledge. This would indeed reduce material and energy flows going along with GDP generation, unless owning many flutes becomes a marker of status, so that people would spend much money on collecting large collections of flutes. Indeed, these flutes accumulate, but they do not only embody value added, but all matterenergy throughputs that are necessary to make them.**

We appreciate that the reviewer feels the result is worth noting. Our argument is that, if it applies for 50 years of civilization growth, it can be reasonably expected to be able to be extrapolated to the future. After all, the "empirical regularity" was first published in 2009 based on 36 years of data, and it has held since to now encompass 50 years.

As for the flute argument, we do not follow. It appears to be conjecture and largely unrelated to the global analysis we perform here. That said, for the sake of argument, flutes, as part of our global culture, do facilitate material and energetic flows

associated with civilization as a whole, a *r*ate that is independent of the *amount* of energy that went into their manufacture, integrals and integrands being orthogonal quantities. An argument could be made that flutes and their performance are valued according to their relative thermodynamic role in civilization, and that flutes will continue to be valued accordingly given their millennia of cultural importance. Because of path-dependence, it is very unlikely that flutes will arbitrarily become much more valuable than everything else in a short time, hence the role for inertia we discuss. Unfortunately, we lack the data to test any flute-related hypothesis.

**Response to the editor**

Minor word-smithing was done throughout the document for style and typos.

As a note in response to an editorial suggestion we have added the text:

*Systems may even undergo quite dramatic changes in character while maintaining at all stages a dependence on previously consumptive states, such as with the succession of species that occurs during development of new forest following a major disturbance (Oliver, 1980)*

We also added an elaboration to the quote by Piketty:

*Like "dark-matter" in astronomy that cannot be seen but is known to be the bulk of our universe, there appears also to be "dark-value" in economics, something that is described well by Piketty, "All wealth creation depends on the social division of labor and on the intellectual capital accumulated over the entire course of human history," continuing "the total value of public and private capital, evaluated in terms of market prices for national accounting purposes, constitutes only a tiny part of what humanity actually values - namely, the part that the community had chosen (rightly or wrongly) to exploit through economic transactions in the marketplace"*

---

## Author Response (AR4)

**I welcome the author's constructive comments. The new diagrams allow me to further specify my critique. But let me start with a simple question. Imagine an economy in which GDP ´shrinks every year with a constant rate, until the economy simply vanishes. According to your approach, this economy would still add a certain amount of GDP every year, hence grow in the absolute terms of cumulative GDP. Would you still stick to your argument? In fact, this economy**
**would also consume less and less energy. In my view, this simple argument shows that your addition approach is deeply flawed.**

It doesn't sound as if the reviewer is questioning the empirical justification of the identity described in the article, which is its main focus. The arguments that are made here follow directly from the identity that is described. However, we agree that interpretation of the result requires some careful consideration, hopefully without immediately concluding that the approach is
flawed. We have rewritten/rearranged some of the text a little to address the reviewer's point.

*We interpret the quantity identified here as the historically cumulative global production $W$ as an economic expression of the rotational power of Lotka's Wheel, that is the capacity to drive the collective to-and-from of civilization's circulations through the relationship $W = wE$, where $w$ is nearly a constant. Certainly, an objection might be raised that the past 50 years is too short relative to the time span of humanity to draw meaningful conclusions about the relationship of historically cumulative*
*production to current energy demands. Measured in units of years, this may be true. However, the last half-century covers a remarkable two-thirds of humanity's total growth expressed in terms of energy consumption, or 1.5 doublings in $E$, during which a great deal changed in humanity's social and technological makeup.*

*Taking the first derivative of Eq. 1 yields an inflation-adjusted economic production relation. Assuming $W = wE$ for constant $w$, then*

$$Y = \frac{dW}{dt} = w\frac{dE}{dt} \tag{1}$$

*Real economic production is related to the* rate of increase *in world primary energy consumption. The implication is that the real GDP is a tally of the instantaneous monetary exchanges that, directly or indirectly, increase civilization's ability to access more energy in the future. For the case that $dY/dt = 0$, namely that there is constant inflation-adjusted economic production $Y$ or zero GDP growth, energy demands expand at rate $Y/w$. If there is GDP growth, as preferred by governments, and*
$d\ln Y/dt > 0$, *then world energy consumption accelerates.*

*Eq. assumes only that $w$ is a constant, a result that can be readily refuted, or supported as it is here with decades of data from multiple sources. The approach does nonetheless have some important limitations, notably an inability to resolve short-term, fine-scale behaviors. The evolution of cumulative inflation-adjusted economic production $W$ is highly smoothed because it is a summation, or integration, over history and the global economy. Even given a strong multi-decadal relationship of $E$ to $W$,*
*year-to-year variability in $E$, such as during recessions or pandemics, cannot be easily related to yearly economic production, especially on national or sectoral scales much smaller than the world as a whole. That said, calculated as a running decadal mean, the average ratio of global production to yearly changes in energy consumption is*

$$\widehat{w} = \frac{Y}{dE/dt} = 5.9 \pm 2.2 \tag{2}$$

*in units of trillion 2019 USD per Exajoule consumed each year, which is very similar to that expressed for $w$ given by Eq. **??**,*
*although the variability is higher given the comparison of $Y$ to a differential in $E$.*

*Nonetheless, Eq. can also be seen as being highly counterintuitive, as it suggests for the hypothetical limiting case of $dE/dt = 0$ – one where the world attains a sort of metabolic steady-state with energetic and material inputs and outputs in balance – that* real *world economic production disappears, that is $Y = 0$. Such a result would seem highly peculiar viewed from any traditional economic perspective.*

*It is important to note, however, that zero real, inflation-adjusted production does not forbid non-zero, positive nominal production. If there is a large difference between the nominal and real GDP, it appears in economic accounts as high values of the GDP deflator, or as hyper-inflation. Interpreted physically, even as current production continues to grow those civilization networks that dissipate energy, there is concurrent rapid fraying of networks constructed through past production, sufficient to offset any productive gains Garrett (2014).*

*Thus, given the severe economic constraints to society, a metabolic steady-state may only represent a temporary marker prior to a more complete collapse, thermodynamic as well as economic. Along this pathway, any external resources that become available to civilization would no longer be sufficient to become an un-utilized residue. Like a patient consumed by cancer, any growth would be more than offset by internal consumption – burning the furniture to heat the house, so to speak. Nominal production may remain, but it is fueled more by internal than external resources. Eventually, the point of complete*
*collapse is reached, whereupon both civilization power and nominal production equal zero.*

**Let me turn to the diagram showing Lotka's wheel. This appears confusing to me. The arrows labelled 'waste' seem to contribute to the growth of the un-utilized residue. That would violate the Second Law, as 'waste' is ultimately entropy production, which by definition cannot be utilized for running the wheel. A correct diagram would show how the wheel contributes to harnessing more energy. Further, the diagram suggests that energy is simply speeding up the wheel. That**
**does not match with Lotka's original account: What is missing is the key role of natural selection. Natural selection is about competing wheels, such that eventually those surpass others in competition that increase energy throughputs, in relative terms.**

We have adjusted the diagram for clarity. The un-utilized residue is not waste, and it contributes as stated in the text to both speeding up and growing the wheel. The aspect of natural selection is not addressed explicitly in this study, although it is a
topic for a future publication. It's not strictly important here as the treatment is of civilization as a whole, which is a) unique in the natural world in terms of its power, and b) doesn't compete with other civilizations.

**Why is this important for assessing the authors' approach? Lotka considers the evolution of structures. The question is how far cumulative annual GDP reflects this. The authors refer to Bettencourt et al. and indeed, the paradigmatic example for the wheel, in the economic context, is urbanization: cities are Lotka's wheels. Urban growth is the key**
**driver of GDP growth and wealth creation. But Bettencourt and many others have argued that the cause of this is precisely what is NOT covered in GDP, by definition, namely positive externalities of knowledge production. This is the gist of New Growth Theory modelling, and central to urban and regional economics. Indeed, positive externalities are what Piketty refers to as 'dark matter'. Knowledge grows, in this sense is a stock, and knowledge guides the designs of technological artefacts that make up Lotka's wheel.**

**Now, if by definition positive externalities are not covered by GDP data, how can one argue that GDP, added up through time, reflects the resulting stock of knowledge? The authors must present a convincing causal account for that, which is still missing. For me, this is simply a logical contradiction, given the definition and measurement of GDP. The authors seem to suggest that cumulative GDP directly reflects network growth in figure 2. But what is the causal process that would establish such a connection?**

We nowhere argue that "the GDP, added up through time, reflects the resulting stock of knowledge". Certainly knowledge (presumably encapsulated in neuronal connections) is one component of civilization networks, but only among a very great many that include such material connections as fiber-optic cables. As for a causal process, this was discussed extensively in prior papers. There is a positive feedback governing the growth of an interface between civilization and its resources (Garrett, 2014, 2015). The past inflation-adjusted GDP built these networks, and these enable the current GDP, so there is a positive
feedback leading to network growth. These dynamics are not the focus of this rather more simple paper which is focused on the constant described, although they are alluded to just prior to the conclusions where we have added the statement.

[Figure]

**Energy and material resources**

**Growth if resources exceed waste**

**Dissipation and decay waste**

**Figure 1.** Representation of Lotka's view on the thermodynamic mechanisms governing system growth, involving a wheel that enlarges and accelerates using an "un-utilized residue" of energy and matter

*In this fashion, the dynamics governing trajectories of civilization's energy consumption and GDP growth can be shown to be governed by resource discovery and environmental decay, and in a manner that accurately reflects its evolution between 1960 and 2010 (Garrett, 2014, 2015).*

**The authors do not need to teach an economics professor 1q'ed. But how does energy consumption causally relate to value added as a measure? I agree that gross production in value terms may also fail to be a good choice, but for that reason I also referred to Material Flow Analysis or other Input-Output frameworks. Every intermediary stage of production is a movement in Lotka's wheel, isn't it? Why do you disregard this as irrelevant?**

    In no way does the paper treat any current action as irrelevant in terms of energy consumption, including the intermediate
steps that factor into gross production. In fact the associated energy consumption is precisely the point. Any current action requires networks only made possible through prior (not current) production, specifically the inflation-adjusted real production that did add to total network value $W$, and hence to the then future capacity to consume energy $E$.

    **Lotka's wheel is not an accounting scheme which ultimately refers to subjective values, meaning preferences as expressed in demand for products priced on the marketplace and thereby becoming manifest in GDP measures. Lotka's**
**wheel is a physical structure moving forward. My flute example is a mere didactic story that illustrates that claiming connection between value added and energy requires a causal account about underlying matterenergy facts. Teaching music could be regarded as extremely valuable, as compared to building fancy cars, in different societies. That would result in very different wheels, wouldn't you agree?**

    **Science does not need to result in universal consensus. Hence, I do not expect you to give up your approach, but to add**
**a convincing story about the underlying physical mechanisms. Perhaps one argument could be that in the past, society preferred the fancy cars, and not the music, hence creating value added resulted in a high powered wheel, through time. That would explain the correlation?**

We believe the reviewer misinterprets the implications of the results. Major cities have symphony orchestras composed of musicians who typically aren't paid that much, and who use human energy to exercise their craft. Meanwhile, the value of their contribution to society is immense – even to those who do not attend performances. Suppose that New York lacked any symphony or opera. It would presumably lose a quantity of real estate value far greater than the amount of money required to simply sustain the orchestra, because the city would be seen generally as being in decline, less culturally vibrant and to many an unattractive place to move. Symphonies are subsidized for good reason. New York without any orchestra would be worth far less.

So simultaneously flutes can consume little energy while teaching flute playing can be highly valued, not because flute teaching is energy intensive but because flutes are embedded as an intrinsic part of a larger network – civilization – that consumes an immense amount of energy. Any given component of society, flutes or fancy cars, cannot be disconnected from any other, at least in any meaningful fashion, nor can be assigned a value uniquely related to its energy consumption without considering the relationship to other things in energy dissipative networks. This is why we focus here on civilization as a whole.

**115 References**

Garrett, T. J.: Long-run evolution of the global economy: 1. Physical basis, Earth's Future, 2, 127–151, https://doi.org/10.1002/2013EF000171, 2014.

Garrett, T. J.: Long-run evolution of the global economy - Part 2: Hindcasts of innovation and growth, Earth Syst. Dyn., https://doi.org/10.5194/esd-6-673-2015, 2015.